

# Enhancing data transmission in duct air quality monitoring using mesh network strategy for LoRa

Amit Mullick[1], Abdul Hadi Abd Rahman[2], Dahlila Putri Dahnil[3] and Nor Mohd Razif Noraini[4]

[1] Faculty of Information Science and Technology, Universiti Kebangsaan Malaysia, Bangi, Selangor, Malaysia
[2] Center for Artificial Intelligence Technology, Universiti Kebangsaan Malaysia, Bangi, Selangor, Malaysia
[3] Center for Software Technology and Management, Universiti Kebangsaan Malaysia, Bangi, Selangor, Malaysia
[4] National Institute of Occupational Safety and Health, Bangi, Selangor, Malaysia

Corresponding author
Abdul Hadi Abd Rahman,
abdulhadi@ukm.edu.my

## ABSTRACT

Duct air quality monitoring (DAQM) is a typical process for building controls, with multiple infections outbreaks reported over time linked with duct system defilement. Various research works have been published with analyses on the air quality inside ducting systems using microcontrollers and low-cost smart sensors instead of conventional meters. However, researchers face problems sending data within limited range and cross-sections inside the duct to the gateway using available wireless technologies, as the transmission is entirely a non-line-of-sight. Therefore, this study developed a new instrument for DAQM to integrate microcontrollers and sensors with a mobile robot using LoRa as the wireless communication medium. The main contribution of this paper is the evaluation of mesh LoRa strategies using our instrument to overcome network disruption problems at the cross-sections and extend the coverage area within the duct environment. A mobile LoRa-based data collection technique is implemented for various data sensors such as DHT22, MQ7, MQ2, MQ135, and DSM50A to identify carbon monoxide, carbon dioxide, smoke, and PM2.5 levels. This study analyzed the efficiency of data transmission and signal strength to cover the air duct environment using several network topologies. The experimental design covered four different scenarios with different configurations in a multi-story building. The network performance evaluations focused on the packet delivery ratio (PDR) and the received signal strength indicator (RSSI). Experimental results in all scenarios showed an improvement in Packet Delivery Ratio (PDR) and significant improvement in the coverage area in the mesh network setup. The results conclude that the transmission efficiency and coverage area are significantly enhanced using the proposed LoRa mesh network and potentially expanded in larger duct environments.

# INTRODUCTION

Today, especially in urban areas, people spend up to 90% of their time indoors. In a fully air-conditioned building with a centralized air-conditioning system, air flows are supplied

to the room through the metal duct channels. The indoor air quality of the building is determined by these duct channels as air circulates inside the occupied range and provides fresh air (*Ibrahim, 2016*). Many infection outbreaks have been reported, which are linked with the contamination of duct systems, cooling towers, ductwork, and filters (*Moscato, Borghini & Teleman, 2017*). So, it is crucial to ensure the supply of fresh and clean air through the duct channels where the ductwork of a building can be contaminated internally in multiple ways (*Liu et al., 2018*). Therefore, periodic air quality for duct channels should be done to maintain the standard air quality and early contamination detection. The conventional method of collecting air samples and analyzing the quality in the laboratory is costly (*Liu, Xia & Zhao, 2016*). Several studies have evaluated and monitored indoor air quality with IoT tools (*Husein, Rahman & Dahnil, 2019*). Research has analyzed DAQM with smart nodes combined with microcontrollers and low-cost IoT sensors instead of commercial meters. The collected data are sent to a remote server using wireless data transmission technologies for the final analysis.

Several technologies are available for wireless communication, such as Bluetooth, Wi-Fi, Zigbee, GiFi, and Wimax (*Garcia et al., 2018*). Previous researches on DAQM used Bluetooth technology between nodes, covering a concise area of wireless data transmission. Some studies used Wi-Fi that shows network disruption at the cross-sections of the duct channel. Data transmission in a duct environment is entirely a non-line-of-sight situation. *Chomba et al. (2011)* showed that Wi-Fi signal strength in a non-line-of-sight indoor environment is reduced to less than $-100$ *dBm* for a 30-m distance between nodes. In a study by *Hashim et al. (2014)*, it was observed that, for outdoor communication, the Wi-Fi signal lost after 150 m. For indoor communication, the signal lost appeared after only 40 m. In indoor situations, the Wi-Fi area coverage decreases due to obstacles in the indoor environment, which reduce the effectiveness of data transmission and result in path loss. Both Wi-Fi and Bluetooth technologies work based on radio wave transmission, and a radio wave cannot pass through metal (*Smith & Smith, 2005*; *Hassan et al., 2016*). In a study by *Swain et al. (2018)*, ZigBee wireless technology sent sensed data from the underground mine to a monitoring station. In this experiment, the researchers experienced packet loss after 135 m and a sudden drop in the signal after 150 m. Path loss for transmitted data is effortless and standard in a non-line-of-sight surrounded environment.

LoRa has emerged as one of the advancements in wireless technology with acceptable receiver sensitiveness and a low amount of bit error rate (BIR). It is considered to have reasonably priced chips for low data rate communication. LoRa can give the longest-range coverage compared with any other current radio technology like Wi-Fi, ZigBee, or Bluetooth (*Daud et al., 2018*). It could cover up to 400m in a non-line of sight environment (*Rahman & Suryanegara, 2017*; *Dahiya, 2017*). However, the coverage area is reduced due to interference of data transmission affected by materials such as that graphite, aluminum foil, steel, and electrically conductive metals that can reflect or even absorb radio waves (*Guan & Chung, 2021*). Based on node amount and connection between nodes, various network topologies have emerged for different usage of LoRa. The most common topology of the LoRa network is Star Topology, Tree Topology, and Mesh Topology. *Tehrani, Amini & Atarodi (2021)* discussed the star and tree topology of the LoRa network, which

is limited to one hop and is defined by the scope of each node. In tree network topology, nodes can act as relays data from a node in a hierarchy farther from the base station in a tree network topology. *Huh & Kim (2019)* state that the LoRa mesh topology model has no hierarchy, unlike in a tree topology. Experimental results showed that the presented method of the LoRa tree network improves the energy consumption of the entire IoT network compared with the star network. Each node can relay a data packet and co-operate with other network nodes to route a packet efficiently into the gateways. Compared to *Lee & Ke (2018)* study, the star and mesh network topologies showed that an increase in communication range by 88.49% PDR, where mesh architecture is an appropriate solution to the issue without installing an additional gateway. Several parameters can be adjusted for different performance targets, like power level, spreading factor, bandwidth, and coding rate. Meanwhile, point-to-point communication-based star topology achieved only 58.7% on average under the same experimental environment. *Hossinuzzaman & Dahnil (2019)* reported an improved network performance significantly a LoRa based mesh network architecture to enhance the packet delivery ratio during rain attenuation. One of the essential features of LoRa technology is Spreading Factor (SF), whereby multiple SF can be used to trade data rate, coverage range of the network, time on the air, receiver sensitivity, longer battery life (*Centenaro et al., 2016*). The drawback of this approach, it could reduce the throughput rate of the network and can be responsible for severe data collision because this setup requires a longer air time for data transmission. This situation appeared due to many LoRa nodes transmitting data and receiving acknowledgments simultaneously (*Lee & Ke, 2018*). These situations can be liable for a massive drop in PDR (*Varsier & Schwoerer, 2017*). For these reasons mentioned above, in our research, we exclude the SF increment to solve the coverage range problem and intend to ensure the best PDR for the network system.

*Abdullah, Leman & Rahman (2013)* developed a mechanical robot that can move through duct channels and collect temperature, humidity, and gas pollutants with sensors and the internal photos of duct channels with a camera. The researchers used a Bluetooth module to transmit the collected data to the data server for final analysis, but the Bluetooth class is unspecified in their research. *Coleman & Meggers (2018)* found that sending the sensed data with Wi-Fi, from inside the duct channels to a remote server outside the duct, is not satisfactory. They acknowledged that in the cross-section of the ventilation air duct, they experienced several problems with Wi-Fi connectivity, which resulted in hardware failure. In order to improve network connectivity, the researchers used an additional antenna, which was an unstable and temporary solution to this problem. In a duct environment, metallic interference of the duct shield reduces signal strength, and a packet cannot reach as far as it should be, limiting the network coverage area. Consequently, when the distance between nodes increases during data transmission, several data packets fail to reach the destination, especially in large buildings or the multilevel of a building. The network coverage area is a significant barrier in gaining the maximum potential of network performance in a non-line-of-sight environment. Therefore, it is vital to select a wireless technology with a vast coverage capacity to overcome communication disruption for transmission of collected data to reduce the packet loss ratio.

This paper proposed a technique for obtaining data from the duct environment to enable air-quality monitoring and secure stable wireless network communication using LoRa with extended area coverage in multi-story buildings with high PDR and strong RSSI. LoRa technology is introduced for DAQM and compared to a two-node-based LoRa point-to-point network architecture. A LoRa based mesh network topology is proposed to cover a large area of DAQM and enhance the wireless communication performance. The main contribution of this paper is the evaluation of mesh LoRa strategies using our instrument to overcome network disruption problems at the cross-sections and extend the coverage area within the duct environment. The remainder of the paper is structured into three sections. The methodology is explained in 'Methodology', followed by the experimental analysis of the proposed objective in 'Experimental Results'. Lastly, the concluding remarks and future works are described in 'Discussions'.

## METHODOLOGY

This section describes the three phases of this research methodology, which consist of node & network architecture, experimental setup, and data collection & evaluation procedures. An Arduino Uno was programmed for data collection with sensors and transmit collected through LoRa in two different network architectures. The master node was set as a data collector in the duct environment and sender the collected data to the destination. The end node was configured as the network receiver and was responsible for visualizing collected data. Two additional repeater nodes were used in mesh network architecture, where each repeater was programmed as a transceiver. Four experimental scenarios were designed to evaluate the network performance in several key parameters.

### Phase 1: node and network architecture

Experiments were conducted based on two network architectures: two node-based networks and a mesh network. In the two node-based networks, one master node was used to sense air elements with sensors in the duct environment and programmed to send the data to the receiver end node through the LoRa wireless technology. Two additional repeater nodes were configured to relay data between the master node and the mesh network's end node.

#### *Two node-based network architecture*

An Arduino Uno microcontroller and one Cytron LoRa RFM shield with a 915MHz antenna acted as primary to achieve master node formation. Five sensors were attached to the board to sense the level of particular elements in the air. The DHT22 sensor was used to measure temperature and humidity, MQ7, MQ2, and MQ135 were used to sense carbon monoxide, smoke, and carbon dioxide levels, respectively, and the DSM501A sensor was used to detect PM2.5 levels. A 5V power bank was used to supply the electricity in the node. The architecture of the network for the point-to-point two node-based communication is illustrated in Fig. 1.

The DJI RoboMaster robot was used to carry all the instruments and travels through the duct channel. The streaming image from the FPV camera helped get visual feedback from the internal part of the ventilation duct. A rechargeable LED torchlight was attached

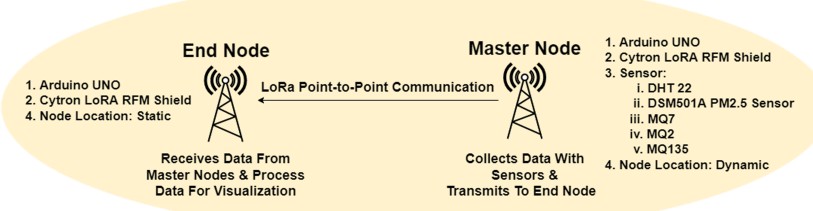

**Figure 1** Two-node-based network architecture.

to the robot's top to light up the pathway inside the duct environment. A Xiaomi Redmi Note 6 Pro android phone was used to operate RoboMaster remotely. An android app named robomaster.apk was installed in the phone to connect with the robot to control the movement of RoboMaster, real-time video monitoring during navigation in duct environment. One end node was programmed to receive the transmitted data from the master node for the sensing sensors. The end node consists of an Arduino Uno board, a Cytron LoRa RFM shield, and a 915-MHz antenna which was configured to display the received data on the serial monitor of the Arduino IDE connected to an Asus X454L laptop, with Arduino IDE application installed for data visualization.

The workflow of nodes and network architecture is defined inside the code of Arduino Uno. The packet length of the data packet in master node equal to 80 bytes (NODE A with ID = 1). The data was routed from the source node to the destination node based on unique Node ID in our network architecture. A Cytron LoRa shield was used master node for the experiment with a maximum of +14 dBm transmission power, TxPower. The master node collected and combined all data, then appended all sensor values as data packets, and finally sent data packets to the destination. All the data packets were formatted as a string, with 60,000 packets are sent serially from the master node. Interval transmission time was set as 2,000 ms between data collection. The end node was configured as the receiver node, defined as NODE D with ID = 4, and 100 bytes of packet length. The data packet receiving policy was configured at end node with calculation of RSSI value. This experiment's network frequency was 915 MHz for both master and end nodes.

The data packet routing policy of the LoRa-based two node-based network at our experiment was that Node A collected data with sensors then sent the data as a packet to Node D based on Node ID. If some other nodes with different node ID or without node ID we present on the nodes, this packet would be discarded due to a mismatch of Node ID. The data packet would be transmitted if Node D was located within the transmission range. After receiving the packet, the end node decoded the data string and visualized received data at the serial monitor, including RSSI value calculated by node D.

*Mesh network architecture*

A mesh network topology was proposed and implemented to improve the data transmission efficiency. In the mesh network between master and end node, two additional nodes were programmed as repeater nodes. Each repeater node consisted of one Arduino Uno, one

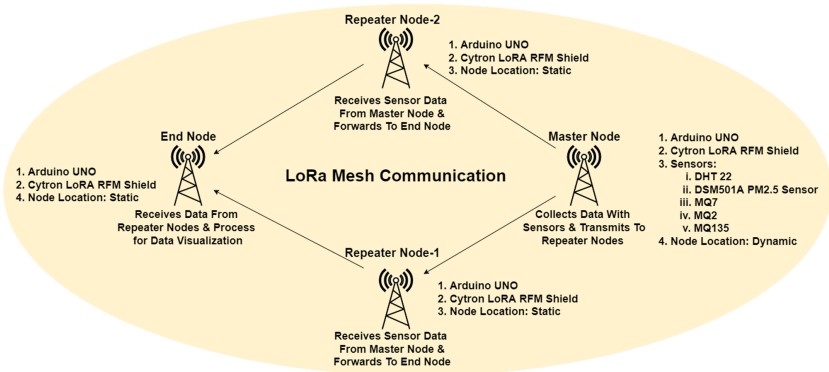

**Figure 2** Mesh network architecture.

Cytron LoRa RFM Shield and 5V power bank. The architecture of the proposed mesh network is presented in Fig. 2.

A master and an end node are used in our LoRa mesh topology. The configuration and workflow of these two nodes in coding was the same as the configuration is used for two-node-based network but the defined destination Node ID at master node and defined source Node ID at receiver end node were different. Repeater nodes are defined as Node B & Node C and the Node ID is defined as respectively 2 & 3. Inside the coding of master node, ID of Node B & C. Data transmitted from Node A will be accepted by Node B & C while discarded other nodes. Repeaters worked as a transceiver to receive data from Node A and forward it to Node D. Then, the repeater node ID was added for the data packet will be transmitted to the end node. Interval time for Node B was set to 3,000 ms, while 1,000 ms for Node C. The network frequency was defined as 915 MHz for all nodes.

Our data packet routing policy used in this study was mesh topology. Node A collected data with sensors and sent it to the repeaters (Node B and Node C) instead of sending it directly to Node D. If both repeaters were within range, the nearest repeater would receive the packet first. The repeater Node ID will be included the packet and sent to Node D. If both repeater nodes receive and transmit the packets, Node D would receive the packet from Node C due to the specific interval settings. Then, the same data packet would be received from Node B. As the string begins with the Node ID, so from the received data string it can be identified that that packet was received from which repeater node, either Node B or Node C. Finally, the RSSI value were calculated and presented *via* the serial monitor. The workflow of nodes in the mesh network is shown in Fig. 3.

## Phase 2: experimental setup

In the experiment, network architecture data transmission was evaluated between different levels on the lab building of the computing department in UKM. In our testbed, the ventilation system consists of two duct channels. One channel is an I-shaped one, supplying fresh air to the air conditioning system. Another duct channel has a zigzag shape, which provides a cool air supply to the rooms. The master node collected data from different points inside the level-one ventilation duct channel, and the end node was placed near

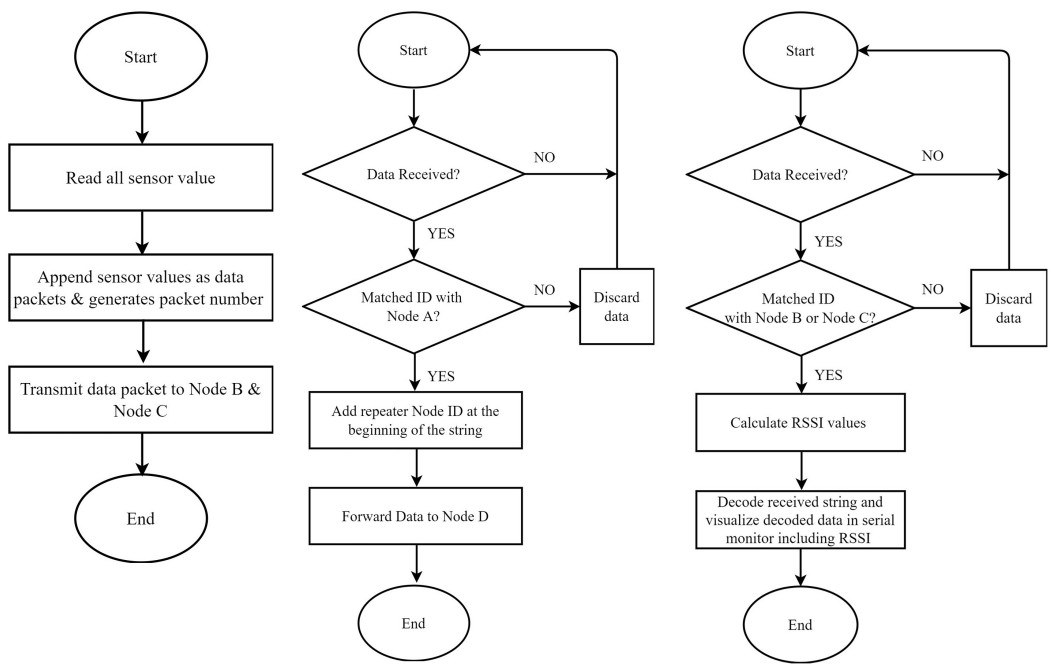

**Figure 3** Flow process of the master node, repeater node, and end node.

the Mechanical Ventilation and Air Conditioning (MVAC) control room of level one. After that, the end node was placed at level two and later at level three. When the master node sends data from different points of the duct channel to the end node, the nature of communication becomes different for each point due to varying distances between the nodes, the different shapes of the duct, and various levels of interference. End-node placement at each level created a different non-line-of-sight network communication situation. In order to evaluate the maximum coverage area of our network, we analyzed data transmission by placing the end node to varying distances inside the faculty compound. The analysis was conducted in four experimental setups to evaluate the data transmission performance. Table 1 presents the descriptions of all four evaluation scenarios used in this study.

For the mesh network, two intermediate repeater nodes are added in parallel height with the master node and two different corners of the outer side of the building to cover the entire area. Figure 4 represents the overview of data transmission analysis, including the figure of all nodes, nine locations of master node placement inside duct map, repeaters, and end node's locations.

Our testbed is a three-story building involving the communication between the master node in level one and the end node in levels two and three. Therefore, the vertical network coverage range was considered based on the experiment results of Scenario-1 and Scenario-2. In the experiment of Scenario-4, the maximum horizontal range of area coverage with a

**Table 1 Overview of the four evaluation scenarios.**

| Setup | Scenario | Description |
|---|---|---|
| Data Transmission Evaluation | Scenario 1—Two-node-based network | The master node collects data to monitor air quality from nine several points inside the duct channel of level one and collected data is sent to the end node for analysis. Points *A, B, C, D, E, F,* and *G* are seven fixed location points for data collection inside the cool air duct channel while points *H* and *I* are fixed location points for the fresh air duct channel. The location of the end node is symbolized with *R*. |
| | Scenario 2—Mesh network | Data transmission is continued following the same procedures as Scenario-1 where the data transmission is conducted using mesh network topology. The collected data from the master node is sent to the repeater nodes. The repeater nodes forward the data to the destination end node. |
| Network Coverage Evaluation | Scenario 3—Two node-based network | The master node is placed in point *N1*, at the approximate center point inside the ventilation duct of level one, and from that point, data is transmitted to the seven points following *O, P, Q, R, S, T & U* which represents the locations of end node placement inside the campus, outside of the lab building. The experiment is conducted here to evaluate how far the master node can transmit data directly to the end node. |
| | Scenario 4—Mesh network | Data transmission has been performed following the same procedures in Scenario-3. The location of the master node and end node placement is also the same as in the previous scenario. Two repeater nodes are placed at locations N1 and N2 in between the master and end node to relay data. The evaluation is performed here with mesh network topology to determine the network coverage area with the highest successful data transmission. |

mesh network was evaluated. Our node placements for the horizontal transmission range test is illustrated in Fig. 5.

## Phase 3: data collection and evaluation procedures

The airflow inside the duct channel is controlled automatically during the experiment. The master node collects data from each point for five minutes, and after the collection, each data packet is immediately sent to the destination end node. The number of packets sent from the master node and the number of packets received by the end node within this timeframe are counted. Based on these parameters, the PDR value was calculated using Eq. (1).

$$\text{Packet Delivery Ratio (PDR)} = \frac{\text{Number of successfully delivered packets}}{\text{Number of Total Transmitted Packets}} * 100 \, (\%). \quad (1)$$

The RSSI value calculation was programmed in the master node and derived from the received data visualization of the end node at Arduino IDE. The entire experiment was repeated three times on three different days to obtain a stable average value. As the experiment was conducted using two different network architectures, the comparison

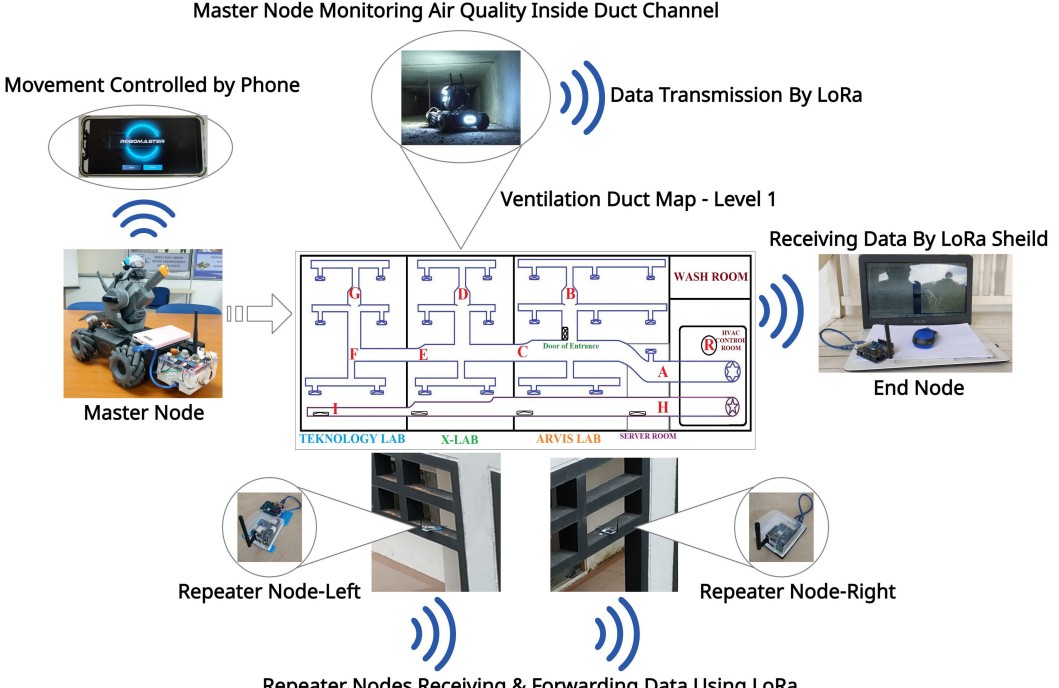

**Figure 4   Overview of data transmission evaluation setup.**

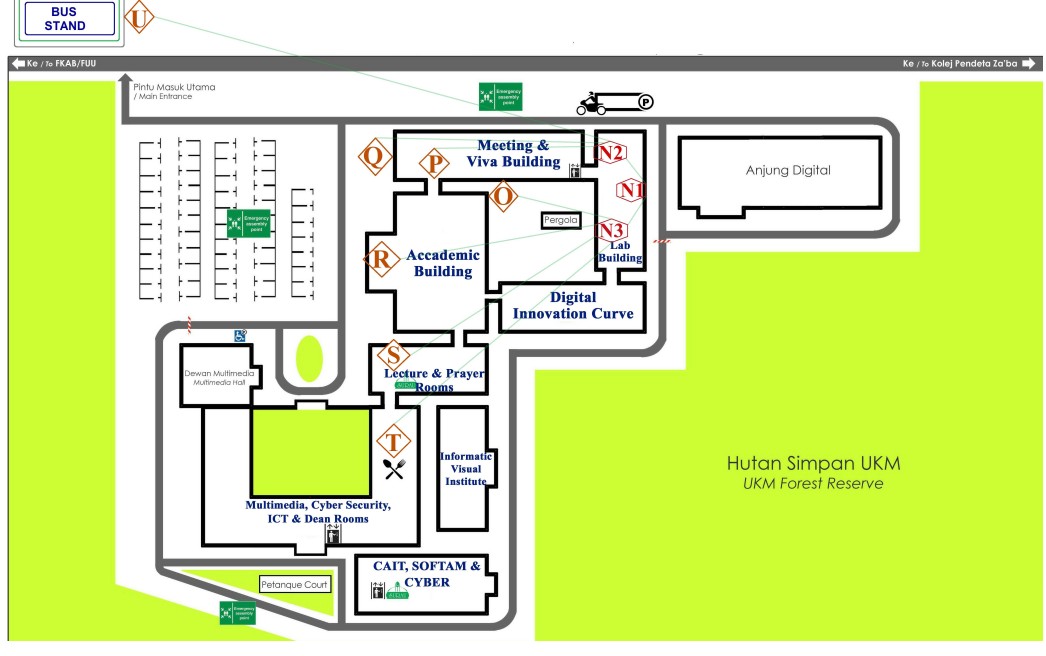

**Figure 5   Overview of network coverage area evaluation.**

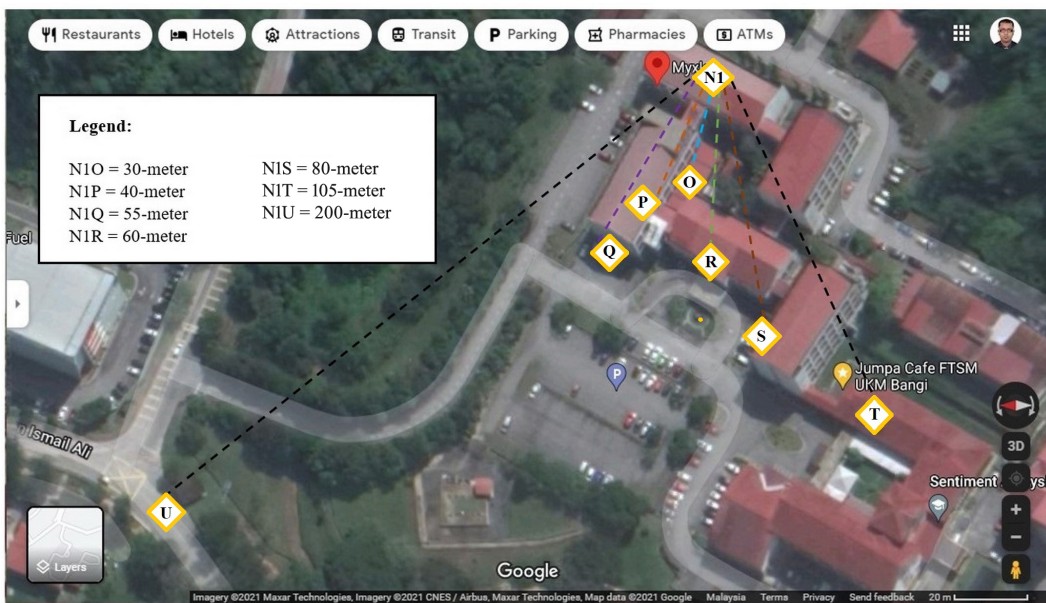

**Figure 6** **Distance between nodes for Scenario 3 and Scenario 4.** Image credit: Imagery @2021 Maxar Technologies, Imagery @2021 CNES/Airbus, Maxar Technologies, Map Data @2021 Google.

between both results was made to calculate the improvement of data transmission using different topologies.

Based on previous research, the distance between the master and end nodes is an important parameter for network performance evaluation. Thus, before starting the experiment, the distance is calculated. In Scenario 3 and Scenario 4, the master node was placed inside the duct channel, and the end node is placed at seven several points in the FTSM campus. Therefore, the distance between nodes for these scenarios extends from the inside of the building to the outside area. For these two scenarios, the distance between the master node location (*N1*) and seven locations of the end node in Google Maps as presented in Fig. 6.

In Scenario 1 and Scenario 2, the master and end nodes were placed in an indoor area, and all the experimental points are situated within the lab building. So, for these two scenarios, the location of both the master and end node was indoors, and it would not be accurate to determine the vertical distance using Google Maps. Since the duct channels are set on top of the room ceiling, it is challenging to directly measure the distance from the master-node location to the end-node location. However, it was possible to measure the entire floor dimension or the distance between any two points on the floor of a building by counting the number of tiles on the floor. From the floor dimension, the distance between the master and end nodes was calculated here using the mathematical-geometric equation of Pythagoras' theorem on right-angled triangles (Eq. (2)):

$$AC^2 = AB^2 + BC^2. \tag{2}$$

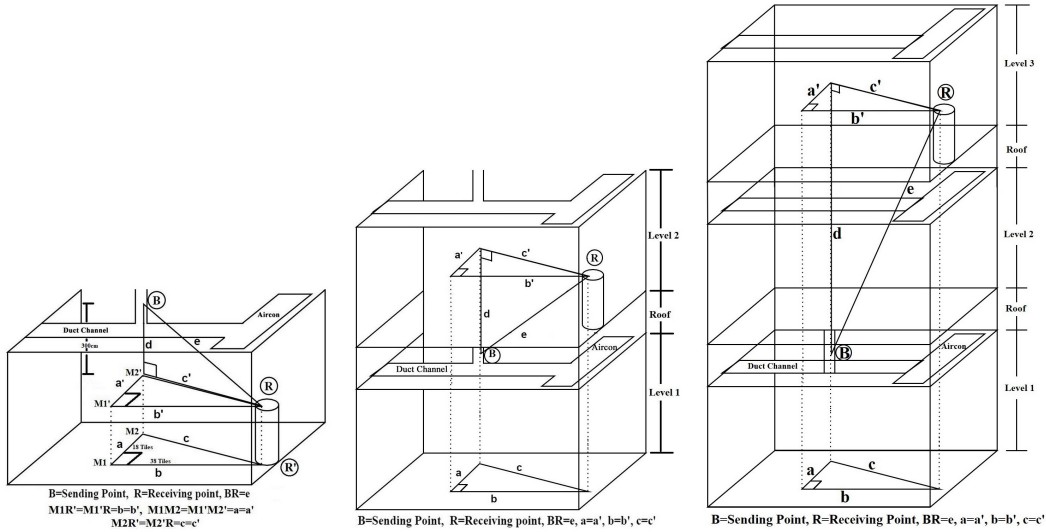

**Figure 7 Distance calculation for level 1, level 2 and level 3.**

Here, ABC is considered a right-angled triangle, and $B = 900$. If the length of AB is measured as "a" and the length of BC is "b" in the above equation, then the AC = c can be determined as in Eq. (3). This equation is known as the hypotenuse equation, derived from Pythagoras' theorem.

$$c = \sqrt{a^2 + b^2}. \tag{3}$$

Two situations are considered which the master and end nodes are both placed at the same level or at two different levels in the building. In this lab building, for level one, the entire floor is 116 tiles in length and 50 tiles in width, and each tile is 30 squared cm in size. So, the floor dimension of each level in the lab building is 3,480 cm x 1,500 cm. The total height of level one from floor to roof is measured at 405 cm, and the roof is 20 cm; the second floor is again 405 cm. Figure 7A presents the distance calculation between the master node location B inside the duct channel and the end node located at point R on top of a chair near the MVAC control room.

R' is the direct vertical point of R, and M2 is the direct vertical point of B at the floor. M1 is the point where a and b are joined in 90° angles. Our measurement shows that; $a = 18$ tiles $= 540$ cm and $b = 38$ tiles $= 1,140$ cm. So M1M2R' is a right-angled triangle. According to Eq. (3), $c = 1,261.43$ cm. M1'M2'R is a mirrored triangle of M1M2R', where a = a', b = b', and c = c'. In this stage, a new triangle M2'RB was imagined, where d is measured with a measurement tape as 300 cm and "c". So, according to Eq. (3), $e = 1,300$ cm (approximately). From point B to end node location R, the distance was approximately 1,300 cm. In the same way, the distance for the other points is also calculated. Following the same formula, the distance were calculated for Level 1—Level 2 and Level 1—Level 3. The calculation methods are illustrated in Figs. 7B and 7C. Various distances between the

**Table 2** Distance between nodes for Scenario-1 and Scenario-2.

| Duct type | Points in Level 1 | Distance to End node in Level 1 | Distance to End node in Level 2 | Distance to End node in Level 3 |
|---|---|---|---|---|
| Cool air duct channel | A | 650 cm | 620 cm | 850 cm |
| | B | 1,300 cm | 1,280 cm | 1,400 cm |
| | C | 1,550 cm | 1,520 cm | 1,650 cm |
| | D | 1,960 cm | 1,950 cm | 2,050 cm |
| | E | 2,250 cm | 2,230 cm | 2,350 cm |
| | F | 2,600 cm | 2,590 cm | 2,700 cm |
| | G | 2,660 cm | 2,640 cm | 2,750 cm |
| Fresh air duct channel | H | 700 cm | 660 cm | 920 cm |
| | I | 2,800 cm | 2,780 cm | 2,870 cm |

master and end nodes for Scenario 1 and Scenario 2 of our experiment are listed in Table 2.

# EXPERIMENTAL RESULTS

This research study involves the experimental evaluation of LoRa data transmission for DAQM and analysis for network area coverage. Data transmission was evaluated in two scenarios, one for the two node-based networks and another for the mesh network. Data transmission was performed at different levels of the building. The network coverage area was analyzed within the computing campus by placing the end node at different locations with different distances from the master node.

## Data transmission

The experiment was first conducted for data transmission evaluation based on a two node-based network architecture, which is considered Scenario 1. The master node was placed at point A and the end node at point R, where the location of both nodes was situated at level one. A total of 23 data packets were generated in the master node between 03:55 PM and 04:00 PM, and each packet is transmitted to the end node immediately after generation. All the received data can be visualized from the Arduino IDE in the end node. All the data packets were successfully received by counting the received packets within that time frame. The maximum RSSI value is recorded for point $A$ at $-87$ and minimum at $-93$. After point $A$, the master node was shifted to point $B$ by driving the RoboMaster. The process continues until the master node reaches point $G$. Then, the master node was taken out of the cold air duct and entered into the straight-shaped fresh air duct line, where points $H$ and $I$ are located. Following the same procedures, three experiment repetitions have been performed for data transmission analysis. After completing the experiment in a two node-based network, it was performed again using the mesh network architecture,

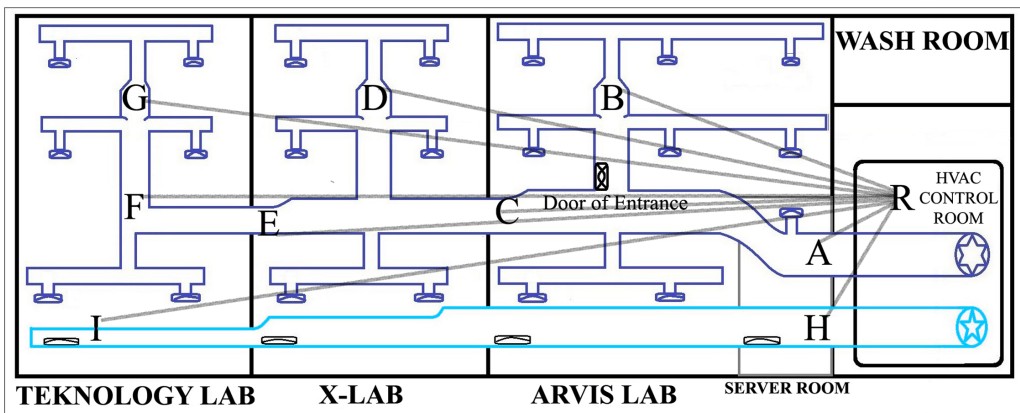

**Figure 8** The master and end node's location at level 1.

which is considered Scenario 2. The location of the master node inside the duct channel and the end node at level one is presented in Fig. 8.

In Fig. 8, points B, D and G were considered as the cross section inside the duct channel for this experiment. The average value of PDR is calculated using the following scenarios: for point *A*, the total number of transmitted data packets in three repetitions was counted as 23 * 3 = 69, and the delivered packets were also counted from the end node as 23 * 3 = 69. Then, PDR is calculated using Eq. (1). Following this formula, the average PDR was calculated for all points from *A* to *I*. For all these points lowest values of RSSI are sorted out. The lowest values from the recorded RSSI were considered here because a low RSSI value is responsible for network disruption and failure of data packet transmission. Points H and I were located inside the fresh air straight-shaped duct channel. For all levels of end node placement, it was found that PDR value for data transmission from H and I is 100% and RSSI value also sufficient strong with two-node-based network architecture although point I was situated at the farthest distance from R. It indicates that, in non-line-of-sight indoor situations, data transmission efficiency depends not only on the distance between nodes but also on the transmission situation. As the cold air duct channel is zigzag-shaped, obstacles and interference are more in there, when straight-shaped fresh air duct channel consists of fewer obstacles comparatively due to its shape. For that reason, all transmission evaluation and comparison value of point H and I were omitted. Figure 9 compares the results between Scenario 1 and Scenario 2 for points inside the cold air duct channel when master and end node both are placed at level 1. From the Fig. 9, it was found that for Scenario 1 and Scenario 2, for all location of master node placement PDR was calculated as 100%. The lowest RSSI for Scenario 1 was recorded as −97 and for Scenario-2 is −88. From the comparison of these results, it was determined that when master node and end node both are placed in level 1, the two node based network was capable of transmitting data from all locations of the master node placement with 100% PDR and strong RSSI.

After experimenting with level 1, the end node was placed at level 2 and the same procedures were repeated. When the master node transmits data from level one to level

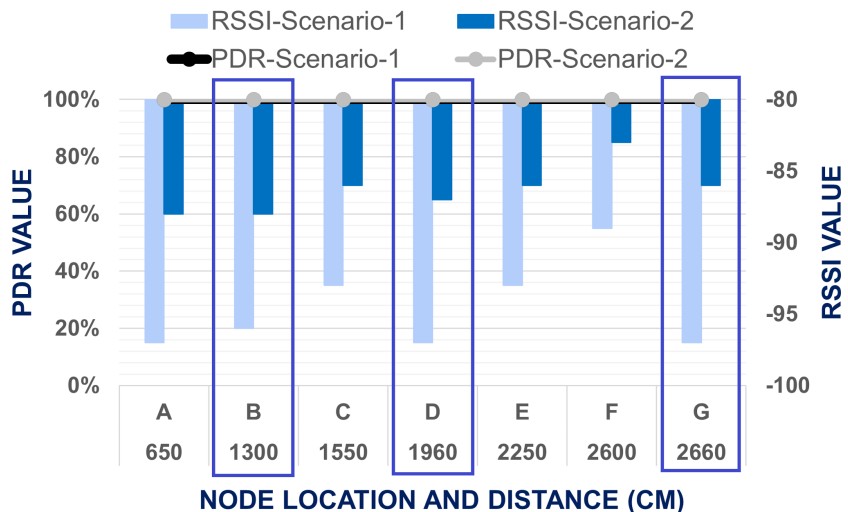

**Figure 9** Evaluation of data transmission when end node placed at level 1.

two using two node-based network architecture, PDR reduces, and RSSI becomes weak. During the data transmission for point A and B, PDR was found 100% but RSSI is reduced compared with level 1. When the master node arrived at point *C*, PDR is reduced to 91.30% and RSSI become −108. Then, at cross points D and G, PDR and RSSI values also decreased compared with level 1. After point *C*, PDR and RSSI were reduced consistently up to point G. This indicates that PDR falls when the distance between the master and end nodes increases. To improve the data transmission efficiency at level 2 same experiment was conducted with mesh network architecture. The comparison of the PDR and RSSI between Scenario 1 and Scenario 2 at level 2 is presented in Fig. 10. It was observed that packets were successfully delivered with 100% PDR and strong RSSI from all points by using mesh network including the cross points. In proposed mesh network lowest RSSI was recorded only −97. At point G where PDR and RSSI was recorded as the lowest accordingly 59.42% and −108 for two node-based network, at same point by using mesh network PDR was founded 100% and RSSI is recorded −91 only. Cross points were not facing any network disruption for data transmission. So, from this results comparison, it was determined that PDR and RSSI is enhanced by using mesh network architecture.

The same experiment is conducted by forwarding the end node at level 3. During data transmission from level one to level three using two node-based network architecture, PDR and RSSI decreased more significantly compared tp level 2. For all data transmission points including cross points network faced packet loss. Lowest PDR is founded at cross point D and point F which was 42.03%, where RSSI is accordingly −109 and −110. In this phase, maximum PDR value was calculated at point A, which is 81.16%. A comparison between Scenario 1 and Scenario 2 when the end node is placed at level 3 is presented at Fig. 11. The proposed mesh network topology can successfully transmit data up to level three with the highest PDR and strong signal. At three cross points B, D and G, PDR was calculated 100% and RSSI is recorded accordingly −94, −96 and −99, which indicates our mesh network

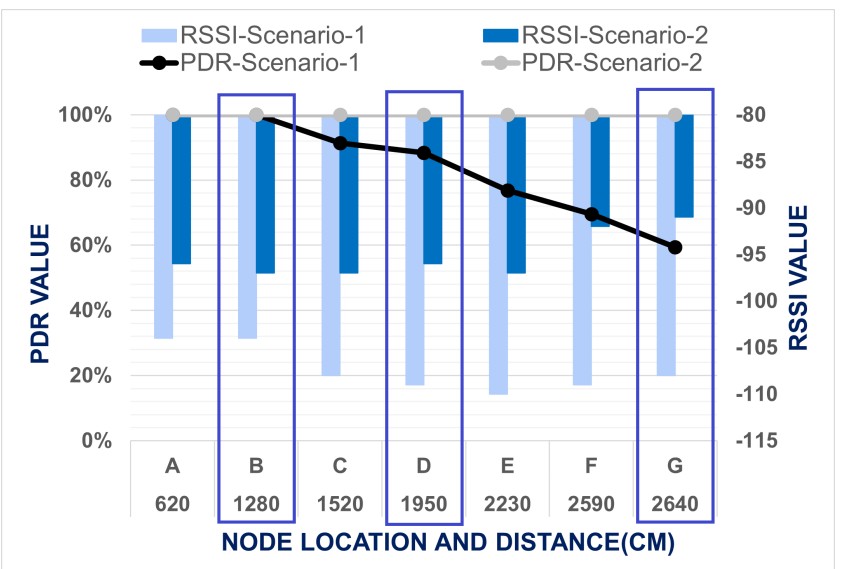

**Figure 10** Evaluation of data transmission when end node placed at level 2.

setup can transmit data to all the cross points too without any network disruptions. PDR was found to be 100% for all points and RSSI between −92 to −100 only. This setup might cover more areas up to level four. However, we limited the scope to test to within three levels to prove that our developed mesh network system was good enough to explore or testbed vertically.

The enhancement of PDR for the mesh network performance is proven from the above-mentioned comparison of the result for Scenario-1 and Scenario-2 at different level of the test bed building. As for level-1, in both scenario, PDR is found 100% and RSSI also bellow −97, so we have compared the result of only level-2 and level-3 to evaluate the improvement of network performance. From the Figs. 10 & 11, it is visualized that the LoRa network is performing comparatively far better in Scenario-2 which is mesh network topology. In Scenario-1, when the end node is placed in level two, only at points *A* & *B* the packet delivery ratio was 100%, but in Scenario-2, in all data collection points, PDR is 100%. Based on the result, the worst situation for Scenario-1 is when the end node was located in level three, and the master node is placed in the cross point *G* at level one. The average PDR is recorded there as 42.03% only. Nevertheless, for the same point of experiment in Scenario-2, the PDR is 100%. So, it was shown that the mesh network improves the data transmission of our system. Based on these worst situations, the improvement of PDR by mesh network was calculated using Eq. (4).

Improvement of PDR

$$= \left( \frac{\text{Recorded PDR in the worst situation with Mesh Network}}{\text{Recorded PDR in the worst situation with Two Node-Based Network}} \right) * 100\%. \quad (4)$$

The improvement of PDR with our proposed mesh network was 237.92%. Improvement also was calculated for RSSI value. It was observed that the lowest RSSI of the experiments

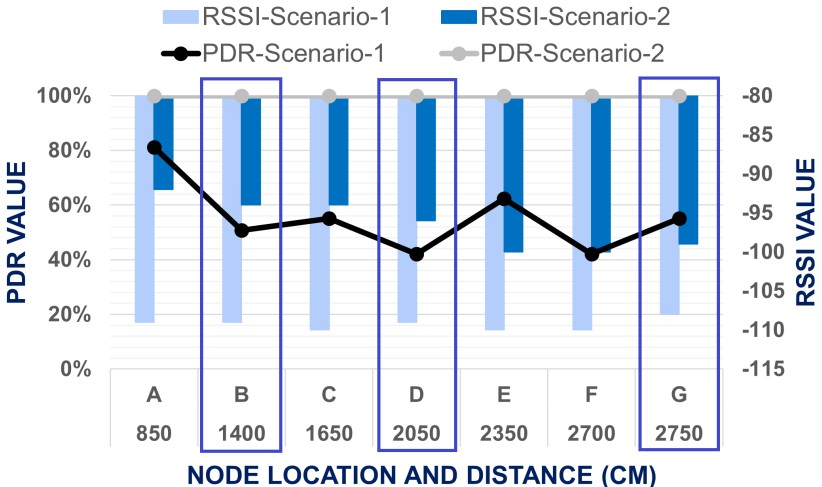

**Figure 11** Evaluation of data transmission when end node placed at level 3.

for two node-based network is recorded −110 when the end node is located in level 3 and master node in Point *G*. For the same point in the mesh network, recorded RSSI was −100 only. The result also shows that when the distance between the master and end nodes is increased, RSSI also decreases. For two node-based network when the end node is placed at level 2, at all data collection points, RSSI value is more than −105 and for level 3, almost all points recorded lowest RSSI value is −110. But in the mesh network for both building levels, RSSI value is recorded from −90 to −100. So based on these worst situations, the improvement of RSSI for the mesh network in Scenario-2 was calculated using Eq. (5). The improvement of RSSI with our proposed mesh network is 10 units. So, it was justified that network efficiency was enhanced in this experiment by using mesh topology.

Improvement of RSSI = (RSSI in the worst situation with Mesh Network

−RSSI in the worst situation with Two Node-Based Network). (5)

## Network coverage area

Furthermore, we expanded our evaluation for coverage area using two node-based networks (Scenario-3) and mesh network (Scenario 4). Data was transmitted from the master node location *N1* to each end node location for 5 min, and the same experiment was repeated three times. The summary of all three repetitions of data collection for the horizontal network coverage test is represented in Fig. 12. The results show that our developed system is eligible for data communication with 100% PDR in our testbed's two nodes-based network setups up to 30 m horizontally. When the end node was shifted to point *P* where the distance between nodes is 40 m, PDR was decreased around 40% and RSSI is minimized down to −109. The end node did not received any signal for the rest of the points Q, R, S, T & U. This severe reduction happened because there was much interference between the master and end node placement points like buildings, pillars, walls, trees, shades, staircase,

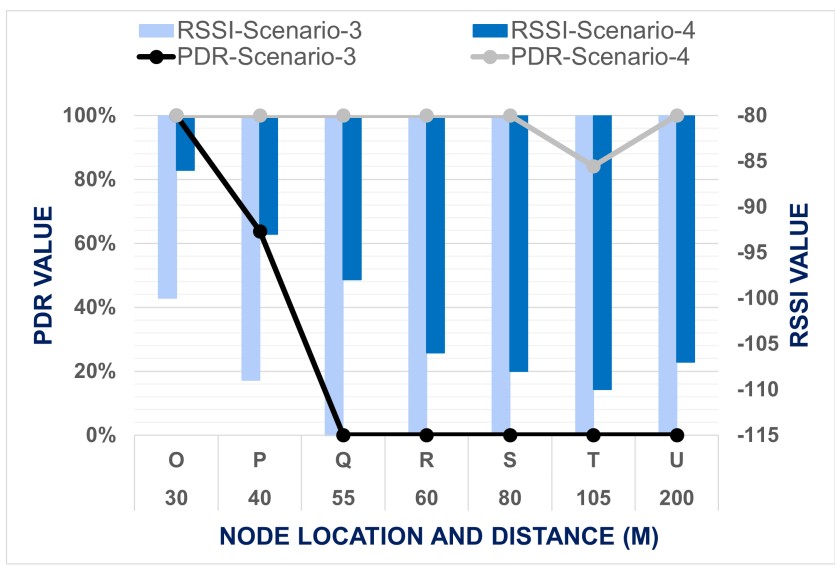

**Figure 12  Comparison between data transmission for Scenario 3 and Scenario 4.**

etc. This interference interrupts the transmitted signal to reach the end node. Therfore horizontally, our developed system cannot transmit data from a far distance.

To cover the entire testbed vertically and monitor the received data from a different campus building, we need to improve the network performance. As an attempt at improvement, a mesh network topology is proposed at this point of the experiment. The mesh network allows all data packets to be successfully transmitted to point *U*, which is 200 m away from the master node. On the other hand, in point *T*, PDR was dropped, although the distance is 105 m here. Figure 6 shows the obstacles existing between the master node location *N1* and end node placement point *U*; there are many trees in between the transmission path that interrupts communications. Still, there is no solid object like wall or building. That is main reason why sent data to point *S* are 100 percent PDR, although this is the point with the highest distance from the master node. On the other hand, from node *N1* to point *T*, there are many walls, buildings, trees in between. As a result, although having less distance comparatively, data transmission is not efficient there, PDR is 84.06%. From Fig. 12, it can be determined that for DAQM of the lab building, the end node can be placed around any block among *A, B, D,* or *E* to transmit data with 100% PDR using the proposed mesh network architecture. It is also observed that mesh topology can comparatively give access within a huge horizontal range. In Scenario-3, the data is transmitted up to 40m distance only in point P with PDR of 63.77%, but in Scenario-4 with mesh network data is transmitted up to 80 m in point T when interference between nodes is more and in 200 m distance at point S when interference between nodes is less. Improvement of network coverage area with the mesh network is calculated by Eq. (6).

Improvement of Coverage Area

$$= \left( \frac{\text{Maximum distance covered with Mesh Network}}{\text{Maximum distance covered with Two Node-Based Network}} \right) * 100\%. \qquad (6)$$

Overall performance on the proposed mesh network showed an improvement of the network coverage area by five times. In Scenario-4 from the master node, data was successfully transmitted to almost all the points of the testbed, while in Scenario-3 data could be transmitted only in two experimental points. It indicates that mesh topology has enhanced the horizontal coverage area of our developed system.

## DISCUSSIONS

The LoRa-based mesh network architecture can transmit data within our testbed area without any kind of network disruption. From the experimentation with different scenarios in the study, it is observed that LoRa technology can successfully transmit received data from the master node to the end node using different network setups. The results show that the two node-based network setup can only cover the area with total efficiency when the master and end nodes are located at level one. PDR and RSSI are decreased with the distance between the master and end nodes. This indicates that the developed platform with the two node-based networks can monitor a small testbed, and it works best within the same level of a building. When the lowest RSSI value becomes −109, the system must experience data loss.

The interference in the transmission path significantly affects data transmission performance over the distance. In Scenario 1, it was seen that transmission performance from point $I$ to end node location $R$ is always better than the other points at less distance. This happens only due to duct shape. Interferences and obstacles are fewer in the transmission path in a straight duct channel than in a zigzag-shaped duct channel. The same result is visible in Scenario 4 if we investigate the data transmission performance from the master node to the end node location point $T$ and point $U$. Data can be transmitted successfully to comparatively far distances if interferences are fewer.

The proposed LoRa-based mesh networks can perform with total efficiency within different levels of testbed buildings. PDR, RSSI, and network area coverage are increased in the mesh network. Improvements from using mesh networks in tested scenarios are; (i) improvement of PDR by two times better and ten units of RSSI in scenarios 1 and 2, and (ii) five-time improvement of horizontal network coverage area in scenarios 3 and 4. The signals become weak quickly in the two node-based networks, obstructed by duct surface and other interferences. So, sometimes, data packets from the master node cannot reach the end node, which results in packet loss. On the contrary, repeater nodes boost weak signals in the mesh architecture. The packets can reach the end node successfully to reduce packet loss. Data packets can travel to the farthest area, resulting in a coverage area extension.

So, this research work provides an improved solution compared with previous research. We introduced LoRa technology for DAQM that can successfully transmit and receive sensor data in non-line-of-sight duct environments. The experimental results showed no significant network disruption at all points in the duct, especially at the cross-sections, with two node-based point-to-point network when the master and end node is placed at the same level of the testbed building. Using the LoRa based mesh network, we significantly

improved network performance both in the horizontal and vertical range, which allows our implemented prototype to explore within a large multi-story building for DAQM. Wireless data transmission is an essential phase of DAQM. There are many obstacles and interferences that exist in a duct environment that affects the network coverage area and reduce transmission efficiency.

## CONCLUSION

This work presents an evaluation of wireless data transmission and enhancement of network performance for DAQM. The study considered several critical scenarios and compared the performance using LoRa based two node-based network architecture and mesh network. The proposed mesh network consists of four nodes and two repeater nodes that can cover the full testbeds of DAQM to transmit sensed data of DAQM to the end node wirelessly for final analysis. The result shows that using a mesh network in this experiment improved PDR at 237% and significantly improved the coverage area by five times compared with a two node-based network. A small mesh network was implemented, including only one master node and two repeater nodes. This study's outcome provides a baseline for implementation in a larger environment considering the influence of the number of nodes. Future studies can explore the integration of an autonomous robot with DAQM to navigate in a duct environment. Furthermore, the visualizing and real-time analysis of the collected data in DAQM can be sent and stored in a cloud system. Another potential area is to utilize using LoRa transmission power, spreading factor (SF), and bandwidth in a larger environment and number of nodes.

### Funding

This works was supported by the Ministry of Higher Education, Malaysia on grant code: FRGS/1/2020/ICT02/UKM/02/7. The funders had no role in study design, data collection and analysis, decision to publish, or preparation of the manuscript.

### Grant Disclosures

The following grant information was disclosed by the authors:
Ministry of Higher Education, Malaysia: FRGS/1/2020/ICT02/UKM/02/7.

### Competing Interests

The authors declare there are no competing interests.

### Author Contributions

- Amit Mullick conceived and designed the experiments, performed the experiments, analyzed the data, performed the computation work, prepared figures and/or tables, and approved the final draft.
- Abdul Hadi Abd Rahman conceived and designed the experiments, performed the experiments, performed the computation work, authored or reviewed drafts of the paper, and approved the final draft.

- Dahlila Putri Dahnil and Nor Mohd Razif Noraini analyzed the data, authored or reviewed drafts of the paper, and approved the final draft.

## Data Availability

The data is available in the Supplemental File and the code is available at GitHub: https://github.com/rdiey84/meshlora.

## Supplemental Information

Supplemental information for this article can be found online at http://dx.doi.org/10.7717/peerj-cs.939#supplemental-information.

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
