# Peer review of "Enhancing data transmission in duct air quality monitoring using mesh network strategy for LoRa"

_PeerJ Computer Science, doi:10.7717/peerj-cs.939_

## Round 0.1 · original submission · Major Revisions

The reviewers have identified some merits of the paper but they also raised some concerns. A thorough revision is needed before further consideration. I look forward to receiving your revised paper.

Reviewer 1 ·

Basic reporting

In the introduction section authors discuss about previous research works related to this topic along with the potential communication technologies for this application. The paper purposes implementation of LoRa for air quality monitoring of large buildings. in this section they should clearly mention what is the novelty and contribution of this paper. In chapter two authors discuss about the previous research work done related to this field. In chapter 3 authors discuss about the experimental setup of this project. In this chapter authors discuss about the experimental setup of the project the. Two communication architecture has been tested. In first architecture end node is directly connected to the master node and in second architecture a repeater node has been used to hop data to the master node. In the results section authors conclude that the mesh network performs better than the direct end-node to master communication on the basis of experimental data.

Experimental design

Two communication architecture has been used
• Direct communication between end-node to master node
• Mesh network-based architecture where an intermediate node has been used for data hopping in case
of weak signal strength.

Authors haven't mentioned what is the uniqueness of their experiment as compared to previous research in this area.

Validity of the findings

Authors have compared the performance of the single hop and two hop based data transmission based on Packet delivery rate for different scenarios. From the experimental results the authors concluded that mesh network-based system provides better
Comments
• Data from table 2-4 can be represented more simply in terms of a chart.
• Figures should be made clearer.
• Use same kind of chart to represent PDR in different scenario in figure 11. Either both histogram of
both lines.
• Novelty not assessed.

Additional comments

• My question is why use LoRa for this specific application? Why not use preexisting communication
infrastructure available in the building like WIFI or LAN?
• Authors should clearly mention what is the nobility and contribution of this paper.
• Authors are suggested to include analysis regarding data rate of the network, Frequency of the data
extraction from each node and how does the size of network effect the data extraction frequency.
• One of the main abilities of LoRa is the ability to transmit data on multiple spreading factor which allows
it to trade data rate and range of communication. So instead of using mesh configuration variable
spreading factors can be used for reliable communication.

Reviewer 2 ·

Basic reporting

1.Some sentences in the text are not clear,for example "The repeater nodes were configured to receive the data and forward it to the end node while the end node was programmed to receive data from the repeater node instead of receiving it directly from the master node."
2.Literature references are sufficient.
3.Figure 1 is not necessary for illustration. Figure 4 is too simple to illustrate the algorithm flow.
4.The experimental results are convincing.
5.Please state the conclusion in more professional terms.

Experimental design

1.The content described in the article is within the objectives and scope of the magazine.
2.Research question well defined, relevant & meaningful. However, the description of the method used is too simple and the description of the network structure is not clear.
3.For the technical description, please specify the function of the mobile phone.
4.The steps in the method description are reasonable and the detailed information is insufficient. For example, the flow chart is too simple.

Validity of the findings

1.The article has certain novelty.
2.The data provided are reliable and sufficient.
3.Please carefully summarize the innovation of the article and write the conclusion in academic words.

Additional comments

1.It is suggested to merge the contents of "2. RELATED WORKS" into "1. INTRODUCTION" because they are all literature analysis.

Reviewer 3 ·

Basic reporting

- Your related works need more detail. I suggest that you consider the other works related to LoRa mesh networks and routing protocols for the wireless mesh networks.
- Figures 4, 5, 6 should be provided in high quality (for example with 600dpi)
- In Figures 11, 12 the comparison of PDRs should be described in 2 columns as the comparison in the Fig. 13.

Experimental design

- In the experiments, the authors added some repeater-nodes between master-nodes and end-node, which organizes multi-hop transmission between them. However, the author should consider the influence of the number of master-nodes and the number of repeater-nodes to PDR.

Validity of the findings

- The results have shown that using additional repeater-nodes increases the PDR and the network coverage in the duct air monitoring application.

Additional comments

- Amit Mullick et al have shown the significant of using a LoRa mesh network for monitoring application, in particularly in duct air monitoring. The authors have proposed a mesh network architecture based on LoRa technology. However, there are repeater-nodes used to relay packets from a master-node to an end-node, and the authors haven’t provided which routing protocols used in the LoRa mesh network.
- The authors should provide more detail about the proposed mesh architecture.

---

## Round 0.2 · accepted · Accept

All reviewers' comments have been addressed. I recommend it for publication.

---

## Author Rebuttal · Round 0.2

UNIVERSITI KEBANGSAAN MALAYSIA
*The National University of Malaysia*

**Pusat Teknologi Kecerdasan Buatan**    *Center for Artificial Intelligence Technology*

Editors,
PeerJ                                                                         1st of March  2022

Dear Editor,

We thank the reviewers for their generous comments on the manuscript "Enhancing data transmission in duct air quality monitoring using mesh network strategy for LoRa".

2. We have edited the manuscript to address their concerns. Here, we provide you with table of corrections for your further evaluation

3. We believe that the manuscript is now suitable for publication in PeerJ Computer Science.

Thank you,

Regards,

Dr Abdul Hadi Abd Rahman
Corresponding author
abdulhadi@ukm.edu.my

PUSAT TEKNOLOGI KECERDASAN BUATAN (CAIT)
Fakulti Teknologi & Sains Maklumat, Universiti Kebangsaan Malaysia, 43600 UKM Bangi, Selangor Darul Ehsan Malaysia
Telefon: +603-8921 6712    Faksimili: +603-8921 6094    E-mel: abdulhadi@ukm.edu.my    Laman Web: http://www.ftsm.ukm.my/cait

Mengilham Harapan, Mencipta Masa Depan • *Inspiring Futures, Nurturing Possibilities*    www.ukm.my

==Response to reviewers==

**Title: Enhancing Data Transmission in Duct Air Monitoring using Mesh Network Strategy for LoRa**

**Manuscript number: 68391**

**Overall Revision Responses**

==We appreciate the time and efforts spent by the editors and reviewers in reviewing our manuscript. The given details are of great benefit to our research. We have considered all the comments and revised the paper accordingly. The revised version is more rigorous and complete. We really appreciate the comments given by the editor and reviewer in ensuring the quality of our paper. Here are the responses to the comments and highlighted in yellow.==

………………………………………………………………………………………………………………………………………………………

# REVIEWER #1

1. **Reviewer #1** *(Basic Reporting - The paper purposes implementation of LoRa for air quality monitoring of large buildings, in this section they should clearly mention what is the novelty and contribution of this paper?*

   ==**Response**: The novelty and contribution are highlighted in the abstract. Page 1. Line 20-22 and in the introduction page 3 Line 123-125.==

   "The main contribution of this paper is the evaluation of mesh LoRa strategies using our instrument to overcome network disruption problems at the cross-sections and extend the coverage area within the duct environment."

2. **Reviewer #1** (Experimental design - *Authors haven't mentioned what is the uniqueness of their experiment as compared to previous research in this area.?*

   **Response**: The uniqueness of the experiments is mentioned in a new added paragraph page 3 line 118-125

   "This paper proposed a technique for obtaining data from the duct environment to enable air-quality monitoring and secure stable wireless network communication using LoRa with extended area coverage in multi-story buildings with high PDR and strong RSSI. LoRa technology is introduced for DAQM and compared to a two-node-based LoRa point-to-point network architecture. A LoRa based mesh network topology is proposed to cover a large area of DAQM and enhance the wireless communication performance. The main contribution of this paper is the evaluation of mesh LoRa strategies using our instrument to overcome network disruption problems at the cross-sections and extend the coverage area within the duct environment."

3. **Reviewer #1** (*Comments - Data from tables 2-4 can be represented more simply in terms of a chart.*)

   **Response**: Yes. The new Figure (9,10,11) are added by replacing tables, and the explanation continues based on that. Page 13, 14 and 15

   "Figure 9, 10 and 11 are updated with the explanations"

[Figure] [Figure] [Figure]

Figure 9: Evaluation of data transmission (End node placed at Level 1)  Figure 10: Evaluation of data transmission (End node placed at Level 2)  Figure 11: Evaluation of data transmission (End node placed at Level 3)

4. **Reviewer #1** (*Comments - Figures should be made clearer?*)

   **Response:** All figures are rebuilt with 600 dpi resolution,

   Figure 1 until 8 (page 4,5,6,8,9,10,11)

5. **Reviewer #1** *(Comments - Use the same kind of chart to represent PDR in different scenario in figure 11. Either both histogram or both lines?*

**Response:** Histogram is used for all scenario to represent PDR and RSSI as shown in new figure 9,10 and 11 on page 13,14,15

"Figure 9, 10 and 11 are updated with the explanations"

"Figure 9 compares the results between Scenario 1 and Scenario 2 for points inside the cold air duct channel when master and end node both are placed at level 1. From the Figure 9, it has been found that for Scenario 1 and Scenario 2, for all location of master node placement PDR is calculated as 100%. The lowest RSSI for Scenario 1 is recorded as -97 and for Scenario-2 is -88. So from these results comparison it is determined that when master node and end node both are placed in level 1, two node based network is capable to transmit data from all location of master node placement with 100% PDR and strong RSSI."

"The comparison of the PDR and RSSI between Scenario 1 and Scenario 2 at level 2 is presented in Figure 10. It is observed that packets are successfully delivered with 100% PDR and strong RSSI from all points by using mesh network including the cross points. In proposed mesh network lowest RSSI is recorded only -97. At point G where PDR and RSSI is recorded as the lowest accordingly 59.42% and -108 for two node-based network, at same point by using mesh network PDR is founded 100% and RSSI is recorded -91only. Cross points are not facing any network disruption for data transmission. So, from this results comparison, it is determined that PDR and RSSI is enhanced by using mesh network architecture."

"In this phase, maximum PDR value is calculated at point A, which is 81.16 %. A comparison between Scenario 1 and Scenario 2 when the end node is placed at level 3 is presented at Figure11. The proposed mesh network topology can successfully transmit data up to level three with the highest PDR and strong signal. At three cross points B, D and G, PDR is calculated 100% and RSSI is recorded accordingly -94, -96 and -99, which

indicates our mesh network setup can transmit data to all the cross points too without any network disruptions. PDR is found 100% for all points and RSSI between -92 to -100 only"

6. **Reviewer #1** *(Comments - Novelty not assessed?)*:

**Response:** New paragraph added on page 17, line 497-504

"So, this research work provides an improved solution compared with previous research. We introduced LoRa technology for DAQM that can successfully transmit and receive sensor data in non-line-of-sight duct environments. Experimental results showed no significant network disruption at all points in the duct, especially at the cross-sections, with two node-based point-to-point network when the master and end node is placed at the same level of the testbed building. Using LoRa based mesh network, we significantly improved network performance both in the horizontal and vertical range, which allows our implemented prototype to explore within a large multi-story building for DAQM. Wireless data transmission is an essential phase of DAQM. There are a lot of obstacles and interferences that exist in a duct environment that affects the network coverage area and reduce transmission efficiency."

7. **Reviewer #1** *(Additional comments- My question is why to use LoRa for this specific application? Why not use preexisting communication infrastructure available in the building like WIFI or LAN?)*

**Response:** The reason of not using preexisting WIFI or LAN is explained on page 2, line 52-55

"Previous researches on DAQM used Bluetooth technology between nodes, covering a concise area of wireless data transmission. Some studies used Wi-Fi that shows network disruption at the cross-sections of the duct channel. Data transmission in a duct environment is entirely a non-line-of-sight situation."

8. **Reviewer #1** (*Additional comments- Authors should clearly mention what is the novelty and contribution of this paper?*)

**Response:** The novelty and contribution are highlighted in the abstract. Page 1. Line 20-22 and in the Introduction, Page 3; Line 123-125

"The main contribution of this paper is the evaluation of mesh LoRa strategies using our instrument to overcome network disruption problems at the cross-sections and extend the coverage area within the duct environment"

9. **Reviewer #1** (*Additional comments- Authors are suggested to include analysis regarding data rate of the network, Frequency of the data extraction from each node and how does the size of network effect the data extraction frequency?*)

**Response:** In this research, we focused on successful data transmission in a non-line-of-sight duct environment. We have selected PDR and RSSI as our performance metrics. The highest transmission power of the LoRa shield is used at the master node to ensure successful transmission. But data rate and bandwidth are related to transmission time, which is another new scope of the study. That's why this point is discussed at the conclusion and mentioned as the future work and advancement of the experiment of this work. Page 17, Line 519-521

"Another potential area is to utilize using LoRa transmission power, spreading factor (SF), and bandwidth in a larger environment and number of nodes."

10. **Reviewer #1** (*Additional comments- One of the main abilities of LoRa is the ability to transmit data on multiple spreading factor which allows it to trade data rate and range of communication. So instead of using mesh configuration variable spreading factors can be used for reliable communication?*)

**Response:** Instead of using multiple SF, mesh networking is chosen in this experiment for the enhancement of network performance. The reason for not using multiple SF is explained on page 3, line 84-96,

"One of the essential features of LoRa technology is Spreading Factor (SF), whereby multiple SF can be used to trade data rate, coverage range of the network, time on the air, receiver sensitivity, longer battery life (Centenaro et al., 2016). The drawback of this approach, it could reduce the throughput rate of the network and can be responsible for severe data collision because this setup requires a longer air time for data transmission."

and the reason for choosing mesh network architecture is explained on page 6, line 195-204

"Our data packet routing policy used in this study is mesh topology. Node A collects data with sensors and sends it to the repeaters (Node B and Node C) instead of sending it directly to Node D. If both repeaters are within range, the nearest repeater will receive the packet first. The repeater Node ID will be included the packet and sent to Node D. If both repeater node receives and transmits the packets, Node D will receive the packet from Node C due to the specific interval settings."

# REVIEWER #2

11. **Reviewer #2** *(Basic Reporting- Some sentences in the text are not clear, for example, "The repeater nodes were configured to receive the data and forward it to the end node while the end node was programmed to receive data from the repeater node instead of receiving it directly from the master node?)*

**Response:** The line of the given example is simplified and rewritten on page 6, line 205-214

"A master and an end node are used in our LoRa mesh topology. The configuration and workflow of these two nodes in coding is the same as the configuration is used for two-node-based network but the defined destination Node ID at master node and defined source Node ID at receiver end node are different. Repeater nodes are defined as NODE B & NODE C and the Node ID is defined as respectively 2 & 3. Inside the coding of master node, ID of Node B & C. Data transmitted from Node A will be accepted by Node B & C while discarded other nodes. Repeaters work as a transceiver to receive data from Node

A and forward it to Node D. Then, the repeater node ID is added for the data packet will be transmitted to the end node. Interval time for Node B is set to 3000 milliseconds, while 1000 milliseconds for Node C. The network frequency is defined as 915 MHz for all nodes."

12. **Reviewer #2** *(Basic Reporting - Figure 1 is not necessary for illustration. Figure 4 is too simple to illustrate the algorithm flow?)*

**Response:** Agreed, Figure-1 is removed from the manuscripts. The algorithm flow is updated. New figure number of the algorithm, Figure-3 on page 6.

Updated Figure 3

[Figure]

Figure 3: Flow chart respectively for the master node, repeater node, and end node

13. **Reviewer #2** *(Basic Reporting - Please state the conclusion in more professional terms?)*
**Response:** Conclusion updated on page 17.

"This work presents an evaluation of wireless data transmission and enhancement of network performance for DAQM. The study considered several critical scenarios and compared the performance using LoRa based two node-based network architecture and mesh network. The proposed mesh network consists of four nodes and two repeater

nodes that can cover the full testbeds of DAQM to transmit sensed data of DAQM to the end node wirelessly for final analysis. The result shows that using a mesh network in this experiment has improved PDR at 237% and significantly improved coverage area by five times compared with a two node-based network. A small mesh network is implemented, including only one master node and two repeater nodes. This study's outcome provides a baseline for implementation in a larger environment considering the influence of the number of nodes."

14. **Reviewer #2** *(Experimental design - However, the description of the method used is too simple and the description of the network structure is not clear?)*

==Response:== The description of the method is updated on Page 4, line 131-139.

"This section describes the three phases of this research methodology, which consist of node & network architecture, experimental setup, and data collection & evaluation procedures. An Arduino Uno is programmed for data collection with sensors and transmit collected through LoRa in two different network architectures. The master node is set as a data collector in the duct environment and sender the collected data to the destination. The end node is configured as the network receiver and is responsible for visualizing collected data. Two additional repeater nodes are used in mesh network architecture, where each repeater is programmed as a transceiver. Four experimental scenarios are designed to evaluate the network performance in several key parameters."

==The network structure is described in detail in new paragraph on page 7, line 220-224.==

"Our data packet routing policy used in this study is mesh topology. Node A collects data with sensors and sends it to the repeaters (Node B and Node C) instead of sending it directly to Node D. If both repeaters are within range, the nearest repeater will receive the packet first. The repeater Node ID will be included the packet and sent to Node D. If both repeater node receives and transmits the packets, Node D will receive the packet from Node C due to the specific interval settings. Then, the same data packet will be received from Node B. As the string begins with the Node ID, so from the received data string it can be identified that that packet is received from which repeater node, either

Node B or Node C. Finally, the RSSI value are calculated and presented via the serial monitor. The workflow of nodes in the mesh network is shown in Figure 3."

15. **Reviewer #2** *(Experimental design - For the technical description, please specify the function of the mobile phone?)*

Response: Function of mobile phone is described on page 4, line 160-163

"A Xiaomi Redmi Note 6 Pro android phone is used to operate RoboMaster remotely. An android app named robomaster.apk is installed in the phone to connect with the robot to control the movement of RoboMaster, real-time video monitoring during navigation in duct environment."

16. **Reviewer #2** *(Experimental design - The steps in the method description are reasonable and the detailed information is insufficient. For example, the flow chart is too simple?)*

Response: Methodology is described in detail from page 4-11, and the flow chart also updated.

Updated methodology section.

[Figure]

**METHODOLOGY**

This section describes the three phases of this research methodology, which consist of node & network architecture, experimental setup, and data collection & evaluation procedures. An Arduino Uno is programmed for data collection with sensors and transmit collected through LoRa in two different network architectures. The master node is set as a data collector in the duct environment and sender the collected data to the destination. The end node is configured as the network receiver and is responsible for visualizing collected data. Two additional repeater nodes are used in mesh network architecture, where each repeater is programmed as a transceiver. Four experimental scenarios are designed to evaluate the network performance in several key parameters.

**Phase 1: Node and Network Architecture**

Experiments are conducted based on two network architectures: two node-based networks and a mesh network. In the two node-based networks, one master node is used to sense air elements with sensors in the duct environment and programmed to send the data to the receiver end node through the LoRa wireless technology. Two additional repeater nodes are configured to relay data between the master node and the mesh network's end node.

*Two node-based network architecture*

An Arduino Uno microcontroller and one Cytron LoRa RFM shield with a 915MHz antenna acted as primary to achieve master node formation. Five sensors are attached to the board to sense the level of particular elements in the air. The DHT22 sensor is used to measure temperature and humidity. MQ7, MQ2, and MQ135 are used to sense carbon monoxide, smoke, and carbon dioxide levels, respectively, and the DSM501A sensor is used to detect PM2.5 levels. A 5V power bank is used to supply the electricity in the node. The architecture of the network for the point-to-point two node-based communication is illustrated in Figure 1.

End Node — Master Node
LoRa Point-to-Point Communication
1. Arduino UNO
2. Cytron LoRa RFM Shield
3. Node Location: Static

Master Node
1. Arduino UNO
2. Cytron LoRa RFM Shield
3. Sensors:
   i. DHT 22
   ii. DSM501A PM2.5 Sensor
   iii. MQ7
   iv. MQ2
   v. MQ135
4. Node Location: Dynamic

Receives Data From Master Node & Process Data For Visualization
Collects Data With Sensors & Transmits To End Node

Figure 1: Two node-based network architecture

DJI RoboMaster robot is used to carry all the instruments and travels through the duct channel. The streaming image from the FPV camera helps get visual feedback from the internal part of the ventilation duct. A rechargeable LED torchlight is attached to the robot's top to light up the pathway inside the duct environment. A Xiaomi Redmi Note 6 Pro android phone is used to operate RoboMaster remotely. An android app named robomaster.apk is installed in the phone to connect with the robot to control the movement of RoboMaster, real-time video monitoring during

navigation in duct environment. One end node is programmed to receive the transmitted data from the master node for the sensing sensors. The end node consists of an Arduino Uno board, a Cytron LoRa RFM shield, and a 915-MHz antenna which configured to display the received data on the serial monitor of the Arduino IDE connected to an Asus X454L laptop, with Arduino IDE application installed for data visualization.

The workflow of nodes and network architecture is defined inside the code of Arduino Uno. The packet length of the data packet in master node equal to 80 bytes(NODE A with ID=1). The data is routed from the source node to the destination node based on unique Node ID in our network architecture. A Cytron LoRa shield is used master node for the experiment with a maximum of +14dBm transmission power, TxPower. The master node collects and combines all data, then append all sensor values as data packets, and finally send data packets to the destination. All the data packets are formatted as a string, with 60,000 packets are sent serially from the master node. Interval transmission time is set as 3000 milliseconds between data collection. The end node is configured as the receiver node, defined as NODE D with ID=4, and 100 bytes of packet length. The data packet receiving policy is configured at end node with calculation of RSSI value. This experiment's network frequency is 915 MHz for both master and end nodes.

The data packet routing policy of the LoRa-based two node-based network at our experiment is that Node A collects data with sensors then sends the data as a packet to Node D based on Node ID. If some other nodes with different node ID or without node ID are present on the nodes, this packet will be discarded due to a mismatch of Node ID. The data packet will be transmitted if Node D is located within the transmission range. After receiving the packet, the end node decodes the data string and visualizes received data at the serial monitor, including RSSI value calculated by node D.

*Mesh network architecture*

A mesh network topology is proposed and implemented to improve the data transmission efficiency. In the mesh network between master and end node, two additional nodes are programmed as repeater nodes. Each repeater node consists of one Arduino Uno, one Cytron LoRa RFM Shield and 5V power bank. The architecture of the proposed mesh network is presented in Figure 2.

End Node — LoRa Mesh Communication — Master Node
Repeater Node 1
Repeater Node 2

Figure 2: Mesh network architecture

A master and an end node are used in our LoRa mesh topology. The configuration and workflow of these two nodes in coding is the same as the configuration is used for two-node-based network but the defined destination Node ID at master node and defined source Node ID at receiver end node are different. Repeater nodes are defined as NODE B & NODE C and the Node ID is defined as respectively 2 & 3. Inside the coding of master node, ID of Node B & C. Data transmitted from Node A will be accepted by Node B & C while discarded other nodes. Repeaters work as a transceiver to receive data from Node A and forward it to Node D. Then, the repeater node ID is added for the data packet will be transmitted to the end node. Interval time for Node B is set to 3000 milliseconds, while 1000 milliseconds for Node C. Only Node B & C IDs. The network frequency is defined as 915 MHz for all nodes.

Our data packet routing policy used in this study is mesh topology. Node A collects data with sensors and sends it to the repeaters (Node B and Node C) instead of sending it directly to Node D. If both repeaters are within range, the nearest repeater will receive the packet first. The repeater Node ID will be included the packet and sent to Node D. If both repeater node receives and transmits the packets, Node D will receive the packet from Node C due to the specific interval settings. Then, the same data packet will be received from Node B. As the string begins with the Node ID, so from the received data string it can be identified that that packet is received from which repeater node, either Node B or Node C. Finally, the RSSI value are calculated and presented via the serial monitor. The workflow of nodes in the mesh network is shown in Figure 3.

Start — Read all sensor value — Append sensor values as data packets & generates packet number — Transmit data packet to Node B & Node C — End

Start — Data Received? — NO / YES — Matched ID with Node A? — YES / NO — Discard data — Add repeater Node ID at the beginning of the string — Forward Data to Node D — End

Start — Data Received? — NO / YES — Matched ID with Node B or Node C? — YES / NO — Discard data — Calculate RSSI value — Decode received string and visualize decoded data in serial monitor including RSSI — End

Figure 3: Flow chart respectively for the master node, repeater node, and end node

**Phase 2: Experimental Setup**

In the experiment, network architecture data transmission is evaluated between different levels on the lab building of the computing department in UKM. In our testbed, the ventilation system consists of two duct channels. One channel is an I-shaped one, supplying fresh air to the air conditioning system. Another duct channel has a zigzag shape, which provides a cool air supply to the rooms. The master node collects data from different points inside the level-one ventilation duct channel, and the end node is placed near the HVAC control room of level one. After that, the end node is placed at level two and later at level three. When the master node sends data from different points of the duct channel to the end node, the nature of communication becomes different for each point due to varying distances between the nodes, the different shapes of the duct, and various levels of interference. End-node placement at each level creates a different non-line-of-sight network communication situation. In order to evaluate the maximum coverage area of our network, we analyze data transmission by placing the end node to varying distances inside the faculty compound. The analysis is conducted in four experimental setups to evaluate the data transmission performance. Table 1 presents the descriptions of all four scenarios.

Table 1: Overview of four scenarios

| Setup | Scenario | Description |
|---|---|---|
| Data Transmission Evaluation | Scenario 1 – Two node-based network | The master node collects data to monitor air quality from nine points inside the duct channel of level one The collected data is sent to the end node for analysis. Points *A, B, C, D, E, F,* and *G* are seven fixed location points for data collection inside the cool air duct channel while points *H* and *I* are fixed location points for the fresh air duct channel. The location of the end node is symbolized with *R*. |
| | Scenario 2 – Mesh network | Data transmission is continued following the same procedures as Scenario-1 where the data transmission is conducted using mesh network topology. The collected data from the master node is sent to the repeater nodes. The repeater nodes forward the data to the destination end node. |
| Network Coverage Evaluation | Scenario 3 – Two node-based network | The master node is placed in point *N1*, at the approximate center point inside the ventilation duct of level one, and from that point, data is transmitted to the seven points following *O, P, Q, R, S, T* & *U* which represents the locations of end node placement inside the campus, outside of the lab building. The experiment is conducted to evaluate how far the master node can transmit data directly to the end node. |
| | Scenario 4 – Mesh network | Data transmission has been performed following the same procedures in Scenario-3. The master node and end node placement location is also the same as in the previous scenario. Besides, two repeater nodes are placed at locations N1 and N2 in between the master and end node to relay data. The evaluation is performed here with mesh network topology to determine the network coverage area with the highest successful data transmission. |

For the mesh network, two intermediate repeater nodes are added in parallel height with the master node and two different corners of the outer side of the building to cover the entire area. Figure 4 represents the overview of data transmission analysis, including the figure of all nodes, nine locations of master node placement inside duct map, repeaters, and end node's locations.

In Scenario 1 and Scenario 2, the master and end nodes are placed in an indoor area, and all the experimental points are situated within the lab building. So, for these two scenarios, the location of both the master and end node is indoors, and it would not be accurate to determine the vertical distance using Google Maps. Since the duct channels are set on top of the room ceiling, it is challenging to directly measure the distance from the master-node location to the end-node location. However, it is possible to measure the entire floor dimension or the distance between any two points on the floor of a building by counting the number of tiles on the floor. From the floor dimension, the distance between the master and end nodes is calculated here using the mathematical-geometric equation of Pythagoras' theorem on right-angled triangles (Equation 2):

$$AC^2 = AB^2 + BC^2 \quad \dots(2)$$

Here, ABC is considered a right-angled triangle, and B = 900. If the length of AB is measured as "a" and the length of BC is "b" in the above equation, then the AC = c can be determined as in Equation 3. This equation is known as the hypotenuse equation, derived from Pythagoras' theorem.

$$c = \sqrt{a^2 + b^2} \quad \dots(3)$$

Two situations are considered which the master and end nodes are both placed at the same level or at two different levels in the building. In this lab building, for level one, the entire floor is 116 tiles in length and 50 tiles in width, and each tile is 30 squared cm in size. So, the floor dimension of each level in the lab building is 3480 cm x 1500 cm. Besides, the total height of level one from floor to roof is measured at 405 cm, and the roof is 20 cm; the second floor is again 405 cm. Figure 7 (a) presents the distance calculation between the master node location B inside the duct channel and the end node located at point R on top of a chair near the HVAC control room.

[Figure]

Figure 7: Distance calculation for (a) L1-L1, (b) L1-L2, and (c) L1-L3

**Phase 3: Data Collection and Evaluation Procedures**

The airflow inside the duct channel is controlled automatically during the experiment. The master node collects data from each point for five minutes, and after the collection, each data packet is immediately sent to the destination end node. The number of packets sent from the master node and the number of packets received by the end node within this timeframe are counted. Based on these parameters, the PDR value is calculated using Equation 1.

$$\text{Packet Delivery Ratio (PDR)} = \frac{Number\ of\ successfully\ delivered\ packets}{Number\ of\ Total\ Transmitted\ Packets} \times 100\ (\%) \quad (1)$$

The RSSI value calculation is programmed in the master node and derived from the received data visualization of the end node at Arduino IDE. The entire experiment is repeated three times on three different days to obtain a stable average value. As the experiment is conducted using two different network architectures, the comparison between both results has been made to calculate the improvement of data transmission using different topologies.

Based on previous research, the distance between the master and end nodes is an important parameter for network performance evaluation. So, before starting the experiment, the distance is calculated. In Scenario 3 and Scenario 4, the master node is placed inside the duct channel, and the end node is placed at seven several points in the FTSM campus. Therefore, the distance between nodes for these scenarios extends from the inside of the building to the outside area. For these two scenarios, the distance between the master node location (*N1*) and seven locations of the end node in Google Maps is presented in Figure 6.

Figure 6: Distance between nodes in satellite view of google map (Scenario-3 & Scenario-4)

R' is the direct vertical point of R, and M2 is the direct vertical point of B at the floor. M1 is the point where a and b are joined in 90° angles. Our measurement shows that; a = 18 tiles = 540 cm and b = 38 tiles = 1140 cm. So M1M2R' is a right-angled triangle. According to Equation 3, c = 1261.43 cm. M1'M2'R is a mirrored triangle of M1M2R', where a = a', b = b', and c = c'. In this stage, a new triangle M2'RB is imagined, where d is measured with a measurement tape as 300 cm and "c". So, according to Equation 3, e = 1300 cm (approximately). So, from point B to end node location R, the distance is approximately 1300 cm. In the same way, the distance for the other points is also calculated. Following the same formula, the distance is calculated for Level 1–Level 2 and Level 1–Level 3. The calculation methods are illustrated in Figures 7(b) and 7(c). Various distances between the master and end nodes for Scenario 1 and Scenario 2 of our experiment are listed in Table 2.

Table 2: Distance between nodes in Scenario-1 & Scenario-2

| Duct type | Point for master node inside duct-Level 1 | Distance when End node in Level 1 | Distance when End node in Level 2 | Distance when End node in Level 3 |
|---|---|---|---|---|
| Cool air duct channel | A | 650cm | 620cm | 850cm |
| | B | 1300cm | 1280cm | 1400cm |
| | C | 1550cm | 1520cm | 1650cm |
| | D | 1960cm | 1950cm | 2050cm |
| | E | 2250cm | 2230cm | 2350cm |
| | F | 2600cm | 2590cm | 2700cm |
| | G | 2660cm | 2640cm | 2750cm |
| Fresh air duct channel | H | 700cm | 660cm | 920cm |
| | I | 2800cm | 2780cm | 2870cm |

**17. Reviewer #2** *(Additional comments - It is suggested to merge the contents of "2. RELATED WORKS" into "1. INTRODUCTION" because they are all literature analysis.?)*

**Response:** Both sections are combined and rephrase. Page 1-3

**INTRODUCTION**

Today, especially in urban areas, people spend up to 90% of their time indoors. In a fully air-conditioned building with a centralized air-conditioning system, air flows are supplied to the room through the metal duct channels. So, the indoor air quality of the building is determined by these duct channels as air circulates inside the occupied range and provides fresh air (Ibrahim, 2016). Many infection outbreaks have been reported, which are linked with the contamination of duct

42 systems, cooling towers, ductwork, and filters (Moscato *et al.*, 2017). So, it is crucial to ensure the
43 supply of fresh and clean air through the duct channels where the ductwork of a building can be
44 contaminated internally in multiple ways (Z. Liu *et al.*, 2018). Therefore, periodic air quality for
45 duct channels should be done to maintain the standard air quality and early contamination detection.
46 The conventional method of collecting air samples and analyzing the quality in the laboratory is
47 costly (S. Liu *et al.*, 2016). Several studies have evaluated and monitored indoor air quality with
48 IoT tools (Husein *et al.*, 2019). Research has analyzed DAQM with smart nodes combined with
49 microcontrollers and low-cost IoT sensors instead of commercial meters. The collected data are
50 sent to a remote server using wireless data transmission technologies for the final analysis.

51       Several technologies are available for wireless communication, such as Bluetooth, Wi-Fi,
52 Zigbee, GiFi, and Wimax (Garcia *et al.*, 2018). Previous researches on DAQM used Bluetooth
53 technology between nodes, covering a concise area of wireless data transmission. Some studies
54 used Wi-Fi that shows network disruption at the cross-sections of the duct channel. Data
55 transmission in a duct environment is entirely a non-line-of-sight situation. (Chomba *et al.*, 2011)
56 showed that Wi-Fi signal strength in a non-line-of-sight indoor environment is reduced to less than
57 –100 *dBm* for a 30-m distance between nodes. In a study by (N. Hashim, N. F. A. M. Azmi1, 2014),
58 it was observed that, for outdoor communication, the Wi-Fi signal lost after 150 meters. For indoor
59 communication, the signal lost appeared after only 40 meters. In indoor situations, the Wi-Fi area
60 coverage decreases due to obstacles in the indoor environment, which reduce the effectiveness of
61 data transmission and result in path loss. Both Wi-Fi and Bluetooth technologies work based on
62 radio wave transmission, and a radio wave cannot pass through metal (Smith & Smith, 2005), and
63 (Hassan *et al.*, 2016). In a study by Swain *et al.*, (2018), ZigBee wireless technology sent sensed
64 data from the underground mine to a monitoring station. In this experiment, the researchers
65 experienced packet loss after 135 m and a sudden drop in the signal after 150 meters. Path loss for
66 transmitted data is effortless and standard in a non-line-of-sight surrounded environment.

67       LoRa has emerged as one of the advancements in wireless technology with acceptable
68 receiver sensitiveness and a low amount of BER (Bit Error Rate). It is considered to have
69 reasonably priced chips for low data rate communication. LoRa can give the longest-range
70 coverage compared with any other current radio technology like Wi-Fi, ZigBee, or Bluetooth
71 (Daud *et al.*, 2018). It could cover up to 400m in a non-line of sight environment (Rahman &
72 Suryanegara, 2017) (Dahiya, 2017). However, the coverage area is reduced due to interference of
73 data transmission affected by materials such as that graphite, aluminum foil, steel, and electrically
74 conductive metals that can reflect or even absorb radio waves (Guan & Chung, 2021). Based on
75 node amount and connection between nodes, various network topologies have emerged for
76 different usage of LoRa. The most common topology of the LoRa network is Star Topology, Tree
77 Topology, and Mesh Topology. Tehrani *et al.*, (2021) discussed the star and tree topology of the
78 LoRa network, which is limited to one hop and is defined by the scope of each node. In tree
79 network topology, nodes can act as relays data from a node in a hierarchy farther from the base
80 station in a tree network topology. Huh & Kim (2019) state that the LoRa mesh topology model
81 has no hierarchy, unlike in a tree topology. Experimental results showed that the presented method
82 of the LoRa tree topology improves the energy consumption of the entire IoT network compared
83 with the star network. Each node can relay a data packet and co-operate with other network nodes
84 to route a packet efficiently into the gateways. Compared to Lee & Ke (2018) study, the star and
85 mesh network topologies showed that an increase in communication range by 88.49% PDR, where
86 mesh architecture is an appropriate solution to the issue without installing an additional gateway.

87 Several parameters can be adjusted for different performance targets, like power level, Spreading
88 Factor, bandwidth, and coding rate. Meanwhile, point-to-point communication-based star
89 topology achieved only 58.7% on average under the same experimental environment.
90 Hossinuzzaman & Dahnil (2019) reported an improved network performance significantly a LoRa
91 based mesh network architecture to enhance the packet delivery ratio during rain attenuation. One
92 of the essential features of LoRa technology is Spreading Factor (SF), whereby multiple SF can
93 be used to trade data rate, coverage range of the network, time on the air, receiver sensitivity,
94 longer battery life (Centenaro *et al.*, 2016). The drawback of this approach, it could reduce the
95 throughput rate of the network and can be responsible for severe data collision because this setup
96 requires a longer air time for data transmission. This situation appeared due to many LoRa nodes
97 transmitting data and receiving acknowledgments simultaneously (Lee & Ke, 2018). These
98 situations can be liable for a massive drop in PDR (Varsier & Schwoerer, 2017). For these reasons
99 mentioned above, in our research, we exclude the SF increment to solve the coverage range
100 problem and intend to ensure the best PDR for the network system.

101      Abdullah *et al.*, (2013) developed a mechanical robot that can move through duct channels
102 and collect temperature, humidity, and gas pollutants with sensors and the internal photos of duct
103 channels with a camera. The researchers used a Bluetooth module to transmit the collected data to
104 the data server for final analysis, but the Bluetooth class is unspecified in their research. Coleman
105 & Meggers (2018) found that sending the sensed data with Wi-Fi, from inside the duct channels
106 to a remote server outside the duct, is not satisfactory. They acknowledged that in the cross-section
107 of the ventilation air duct, they experienced several problems with Wi-Fi connectivity, which
108 resulted in hardware failure. In order to improve network connectivity, the researchers used an
109 additional antenna, which was an unstable and temporary solution to this problem. In a duct
110 environment, metallic interference of the duct shield reduces signal strength, and a packet cannot
111 reach as far as it should be, limiting the network coverage area. Consequently, when the distance
112 between nodes increases during data transmission, several data packets fail to reach the destination,
113 especially in large buildings or the multilevel of a building. The network coverage area is a
114 significant barrier in gaining the maximum potential of network performance in a non-line-of-sight
115 environment. Therefore, it is vital to select a wireless technology with a vast coverage capacity to
116 overcome communication disruption for transmission of collected data to reduce the packet loss
117 ratio.

118      This paper proposed a technique for obtaining data from the duct environment to enable
119 air-quality monitoring and secure stable wireless network communication using LoRa, with
120 extended area coverage in multi-story buildings with high PDR and strong RSSI. LoRa technology
121 is introduced for DAQM and compared to a two-node-based LoRa point-to-point network
122 architecture. A LoRa based mesh network topology is proposed to cover a large area of DAQM
123 and enhance the wireless communication performance. The main contribution of this paper is the
124 evaluation of mesh LoRa strategies using our instrument to overcome network disruption problems
125 at the cross-sections and extend the coverage area within the duct environment. The remainder of
126 the paper is structured into three sections. The methodology is explained in Section 2, followed by
127 The experimental analysis of the proposed objective in Section 3. Lastly, the concluding remarks
128 and future works are described in Section 4.

# REVIEWER #3

**18. Reviewer #3** *(Basic Reporting - Your related works need more detail. I suggest that you consider the other works related to LoRa mesh networks and routing protocols for the wireless mesh networks?)*

**Response:** New paragraph is added on page 2, line 55-66

"(Chomba et al., 2011) showed that Wi-Fi signal strength in a non-line-of-sight indoor environment is reduced to less than –100 dBm for a 30-m distance between nodes. In a study by (N. Hashim, N. F. A. M. Azmi1, 2014), it was observed that, for outdoor communication, the Wi-Fi signal lost after 150 meters. For indoor communication, the signal lost appeared after only 40 meters. In indoor situations, the Wi-Fi area coverage decreases due to obstacles in the indoor environment, which reduce the effectiveness of data transmission and result in path loss. Both Wi-Fi and Bluetooth technologies work based on radio wave transmission, and a radio wave cannot pass through metal (Smith & Smith, 2005), and (Hassan et al., 2016). In a study by Swain et al., (2018), ZigBee wireless technology sent sensed data from the underground mine to a monitoring station. In this experiment, the researchers experienced packet loss after 135 m and a sudden drop in

the signal after 150 meters. Path loss for transmitted data is effortless and standard in a non-line-of-sight surrounded environment."

**19.** **Reviewer #3** *(Basic Reporting - Figures 4, 5, 6 should be provided in high quality for example with 600dpi?)*

**Response:** Figure 4,5 and 6 are rebuilt with 600 dpi resolution

[Figure]

Figure 4: Overview of data transmission analysis

Figure 5: Overview of network coverage area analysis

Figure 6: Distance between nodes in satellite view of google map
(Scenario-3 & Scenario-4)

**20.** **Reviewer #3** *(Basic Reporting - In Figures 11, 12 the comparison of PDRs should be described in 2 columns as the comparison in the Fig. 13.?)*

**Response:** The figures are updated and updated explanation pattern for each figure. Page 13-16, line 405-515

"Figure 9, 10 and 11 are updated with the explanations"

[Figure]

Figure 9: Evaluation of data transmission (End node placed at Level 1)

[Figure]

Figure 10: Evaluation of data transmission (End node placed at Level 2)

[Figure]

Figure 11: Evaluation of data transmission (End node placed at Level 3)

**21. Reviewer #3** *(Experimental design - However, the author should consider the influence of the number of master-nodes and the number of repeater-nodes to PDR.?)*

**Response:** This issue is considered as the limitation of our work and explained inside the conclusion part on page 17, line 519-521.

"Another potential area is to utilize using LoRa transmission power, spreading factor (SF), and bandwidth in a larger environment and number of nodes."

**22. Reviewer #3** *(Additional comments- and the authors haven't provided which routing protocols used in the LoRa mesh network. - The authors should provide more detail about the proposed mesh architecture?)*

**Response:** The data routing policy for both network architecture is included, and detail description of proposed mesh network architecture is added on page 5-6, line 187-198

"A mesh network topology is proposed and implemented to improve the data transmission efficiency. In the mesh network between master and end node, two additional nodes are programmed as repeater nodes. Each repeater node consists of one Arduino Uno, one Cytron LoRa RFM Shield and 5V power bank. The architecture of the proposed mesh network is presented in Figure 2"

-END OF DOCUMENT-